# *Tm*Spz-like Plays a Fundamental Role in Response to *E. coli* but Not *S. aureus* or *C. albican* Infection in *Tenebrio molitor* via Regulation of Antimicrobial Peptide Production

**DOI:** 10.3390/ijms221910888

**Published:** 2021-10-08

**Authors:** Ho Am Jang, Bharat Bhusan Patnaik, Maryam Ali Mohammadie Kojour, Bo Bae Kim, Young Min Bae, Ki Beom Park, Yong Seok Lee, Yong Hun Jo, Yeon Soo Han

**Affiliations:** 1Department of Applied Biology, Institute of Environmentally-Friendly Agriculture (IEFA), College of Agriculture and Life Sciences, Chonnam National University, Gwangju 61186, Korea; hoamjang@gmail.com (H.A.J.); maryam.alimohammadie@gmail.com (M.A.M.K.); kbb941013@gmail.com (B.B.K.); ugisaka@naver.com (Y.M.B.); misson112@naver.com (K.B.P.); 2P.G. Department of Biosciences and Biotechnology, Fakir Mohan University, Balasore, Odisha 756089, India; drbharatbhusan4@gmail.com; 3Department of Biology, College of Natural Sciences, Soonchunhyang University, Asan City 31538, Korea; yslee@sch.ac.kr

**Keywords:** Spätzle, *T. molitor*, RNA interference, antimicrobial peptide, NF-κB, innate immunity

## Abstract

The cystine knot protein Spätzle is a Toll receptor ligand that modulates the intracellular signaling cascade involved in the nuclear factor kappa B (NF-κB)-mediated regulation of antimicrobial peptide (AMP)-encoding genes. Spätzle-mediated activation of the Toll pathway is critical for the innate immune responses of insects against Gram-positive bacteria and fungi. In this study, the open reading frame (ORF) sequence of *Spätzle-like* from *T. molitor* (*TmSpz-like*) identified from the RNA sequencing dataset was cloned and sequenced. The 885-bp *TmSpz-like* ORF encoded a polypeptide of 294 amino acid residues. *Tm*Spz-like comprised a cystine knot domain with six conserved cysteine residues that formed three disulfide bonds. Additionally, *Tm*Spz-like exhibited the highest amino acid sequence similarity with *T. castaneum* Spätzle (*Tc*Spz). In the phylogenetic tree, *Tm*Spz-like and *Tc*Spz were located within a single cluster. The expression of *TmSpz-like* was upregulated in the Malpighian tubules and gut tissues of *T. molitor*. Additionally, the expression of *TmSpz-like* in the whole body and gut of the larvae was upregulated at 24 h post-*E. coli* infection. The results of RNA interference experiments revealed that *TmSpz-like* is critical for the viability of *E. coli*-infected *T. molitor* larvae. Eleven AMP-encoding genes were downregulated in the *E. coli*-infected *TmSpz-like* knockdown larvae, which suggested that *TmSpz-like* positively regulated these genes. Additionally, the NF-κB-encoding genes (*TmDorX1*, *TmDorX2*, and *TmRelish*) were downregulated in the *E. coli*-infected *TmSpz-like* knockdown larvae. Thus, *TmSpz-like* plays a critical role in the regulation of AMP production in *T. molitor* in response to *E. coli* infection.

## 1. Introduction

As invertebrates lack adaptive immunity, their immune responses against pathogens solely depend on innate immunity and physical barriers, such as chitinous shells. The innate immune response is mediated through cell-mediated mechanisms such as encapsulation, nodulation, and phagocytosis or the humoral responses that lead to the production of antimicrobial peptides (AMPs). AMP production, which is the most conserved immune effector mechanism in invertebrates, is induced through the activation of the Toll and IMD signaling cascades. The pattern recognition receptors (PRRs) of the signaling cascades identify the non-self carbohydrate moieties enveloping the pathogens, which are commonly called pathogen-associated molecular patterns (PAMPs). The interaction between PRRs and PAMPs, which can be specific or non-specific, promotes signal transduction through the Toll or IMD proteins. The Toll-like receptor (TLR)-nuclear factor kappa B (NF-κB) pathway has been elucidated in *Drosophila*. In this signaling pathway, the PRRs recognize PAMPs, such as lipopolysaccharides (LPS), peptidoglycans (PGN), and β-glucans, on the surface of bacteria and fungi and trigger a multi-step proteolytic cascade of serine proteases. Modular serine protease (MSP), which activates the Toll ligand Spätzle (Spz) by processing the inactive pro-Spz, triggers the intracellular signaling cascade mechanism. The binding of Spz to Toll results in the recruitment of the adaptor proteins MyD88, Tube, and Pelle and consequently modulates the responses of the NF-κB factor Dorsal. The cascade mechanism then activates the transcription factor NF-κB and upregulates the expression of the effector genes encoding AMPs [1]. The serine protease cascade model has been proposed in *Tenebrio molitor* [2,3]. *Tenebrio* GNBP1/PGRP-SA and GNBP3 bind to Lys-type PGN and β-1,3-glucans, respectively, of bacterial and fungal cell walls, and consequently, in the presence of Ca^2+^ ions, recruit pro-MSP and process it into activated MSP (aMSP). The aMSP eventually processes pro-Spz to mature Spz along with activated Spz processing enzyme (SPE). SPE itself is recruited by SPE-activating enzyme (SAE). Activated Spz promotes the production of AMPs such as Tenecin 1 and Tenecin 2 through the Toll receptor. Briefly, the TLR-NF-κB pathway in *T. molitor* is dependent on the cleaved form of Spz. This suggests that Spz proteins have promiscuous roles in the innate immunity of insects against pathogens [4]. Furthermore, *Tm*Spz proteins can function as central immune modulators for biosurfactant-mediated AMP production in *T. molitor* [5].

In *Bombyx mori*, the Toll-Spz pathway is critical for eliciting an immune response. Previous studies have reported that *Bm*Toll11, *Bm*Toll9–1, and five *BmSpz* genes are upregulated after challenge with *Escherichia coli* and *Staphylococcus aureus*. Additionally, only *Bm*Spz2 can interact with *Bm*Toll 11 and *Bm*Toll9–1 and activate AMP-encoding gene expression [6]. Meanwhile, *Bm*Spz4 is reported to be involved in integument AMP production after challenge with *Bacillus* and yeast [7]. In *Rhynchophorus ferrugineus* (red palm weevil), the Spz homolog-mediated Toll-like pathway activation regulates AMP response and maintains gut homeostasis. AMPs such as *R. ferrugineus Coleoptericin* (*RfColeoptericin*) and *RfCecropin* are downregulated in *RfSpz* knockdown insects, which suggested that their secretion is regulated by *Rf*Spätzle-mediated activation of the Toll signaling pathway [8]. The larvae of *T. molitor*
*Spätzle-4* (*TmSpz-4*) knockdown insects are susceptible to *E. coli* and *C. albicans* infections due to the downregulation of AMPs and Toll pathway-related NF-κB factors (Dorsal X1 and Dorsal X2) [9,10]. Similar results were observed with *TmSpz-6* knockdown larvae with positive regulation of AMPs (*TmTenecin-2* and *TmTenecin-3*) after challenge with *E. coli* and *S. aureus* [11]. In *Drosophila*, multiple Toll-Spz combinations activate the antifungal peptide-encoding gene *Drosomycin*. *Drosophila* Toll-1 and Toll-7 ectodomains bind to Spz-1, Spz-2, and Spz-5 and activate the expression of *Drosomycin* [12,13]. Spz homologs and their conserved functions in the Toll pathway have been elucidated in other insects, such as *Manduca sexta* [14], *Antheraea pernyi* [15], *Anopheles gambiae* [16], *Aedes aegypti* [17], as well as in the shrimps *Litopenaeus vannamei* [18,19] and *Fenneropenaeus chinensis* [20].

Previously, we have analyzed the RNA sequencing database of *T. molitor* to screen various innate immunity-related transcripts [10,21,22,23]. Nine different *TmSpätzle* (*TmSpz*) gene sequences were identified from the database using the *T. castaneum* Spz sequence (XP_008201191.1) as the query. In this study, a signal peptide region, a cleavage site, and a cysteine knot domain were identified in the novel Spätzle-like from *T. molitor* (*Tm*Spz-like). The expression of *TmSpz-like* transcripts was examined during development and in various tissues of the insect using quantitative real-time polymerase chain reaction (qRT-PCR). The expression of *TmSpz-like* transcripts was upregulated after challenge with microorganisms. RNA interference (RNAi) studies were performed to examine the effect of *TmSpz-like* knockdown on the viability of the larvae challenged with microorganisms. The findings of this study indicated that the viability of *TmSpz-like* knockdown larvae infected with *E. coli* was correlated with the levels of AMP-encoding transcripts. Among the Toll signaling downstream NF-κB factors, the levels of *T. molitor Dorsal isoforms* (*X1* and *X2*) were downregulated after infection with Gram-negative bacteria, Gram-positive bacteria, and a fungus. Only Gram-negative bacteria downregulated the levels of *Relish*, a downstream transcription factor in IMD signaling. These findings indicate that the *TmSpz-like* transcript is involved in the innate immunity of *T. molitor*.

## 2. Results

### 2.1. Cloning and Sequence Analysis of TmSpz-like

The full-length ORF sequence of *TmSpz-like* (Accession no.: MZ708792) was screened from the RNAseq database using the *T. castaneum Spz* (*TcSpz*) sequence (XP_008201191.1) as a query in the tblastn analysis. Molecular cloning analysis confirmed and validated the nucleotide and deduced amino acid sequences. The ORF region of *TmSpz-like* was 885 bp in length and encoded a protein with 294 amino acid residues. *Tm*Spz-like protein includes a signal peptide region at the N-terminus (cleaved after the first 25 amino acid residues) and a cysteine knot region (amino acid residues 168–269) in the C-terminus containing conserved cysteine residues (residues 175, 215, 222, 236, 265, 266, and 267) that can potentially bind to the Toll receptor (Figure 1). The molecular weight and theoretical isoelectric point of the putative protein were 33.79 kDa and 8.82, respectively. Pro-Spz is cleaved by SPE. *Tm*Spz-like protein exhibited the characteristics of the Spz family members as evidenced by the presence of cleavage sites.

### 2.2. Phylogenetic Analysis of TmSpz-like

Multiple sequence alignment was performed to investigate the genetic relationship of *Tm*Spz-like with its homologs. Conserved cysteine residues located in the Spz domain were involved in disulfide bond formation, while one conserved cysteine residue was involved in dimerization. *Tm*Spz-like exhibited the highest amino acid sequence similarity with *Tc*Spz (85%), followed by *Bombus impatiens* Spz (*Bi*Spz; 49%) and *Neodiprion lecontei* Spz (*Nl*Spz), *Cimex lectularius* Spz (*Cl*Spz), and *Halyomorpha halys* Spz (*Hh*Spz) (43% for all) (Appendix A). As shown in Figure 2, *Tm*Spz-like was located closed to its coleopteran counterpart *Tc*Spz in the phylogenetic tree, while *Bi*Spz, *Ar*Spz-like, and *Nl*Spz (Hemipterans) comprised another taxon in the same cluster. The Spz sequences belonging to the orders Lepidoptera, Diptera, and Hymenoptera clustered together on separate branches. Spz5-like of the decapod crustacean *P. vannamei* segregated as an outgroup in the phylogenetic tree. Additionally, the 3D simulation of *Tm*Spz-like (exhibiting 34.31% identity with the reference crystal structure of Spz cysteine knot homodimer) confirmed the location of two *Tm*Spz-like proteins and determined dimerization and disulfide bond positions of cysteine residues (Appendix A).

### 2.3. Developmental and Tissue-Specific Expression Patterns of TmSpz-like

Developmental and tissue-specific expression patterns of *TmSpz-like* mRNA were analyzed using qRT-PCR (Figure 3). *TmSpz-like* mRNA expression was detected in all developmental stages and tissues, with the highest expression detected in the egg and young larval stages (Figure 3A). The expression of *TmSpz-like* mRNA decreased from the young larval to the pre-pupal stage and subsequently increased on day 1 of the pupal stage and decreased again by day 7. Compared with those in the pupal stages, the *TmSpz-like* mRNA levels were upregulated in the adult stage. The tissue distributions of *TmSpz-like* mRNA in *T. molitor* larva and adults are shown in Figure 3B,C, respectively. qRT-PCR analysis revealed that *TmSpz-like* mRNA is mostly expressed in the Malpighian tubules of the late larvae. The expression of *TmSpz-like* mRNA in the Malpighian tubules was upregulated by 15-fold compared with that in other tissues (Figure 3B). In adults, the expression of *TmSpz-like* mRNA in the gut was upregulated by 20-fold compared with that in other tissues (Figure 3C).

### 2.4. Effect of Microbial Infection on the Expression Patterns of TmSpz-like

Next, the effect of immune elicitors on the expression of *TmSpz-like* was examined. The expression levels of *TmSpz-like* mRNA were examined at 3, 6, 9, 12, and 24 h in different tissues (gut, fat body, hemocytes, and Malpighian tubules) and the whole body after infection with *E. coli*, *S. aureus*, and *C. albicans* (Figure 4). To account for the effect of injection during the experiment, PBS (pH 7)-injected larvae were used as a negative control. The expression of *TmSpz*-like mRNA was mostly detected in the whole body at 24 h post-*E. coli* injection (Figure 4A). Additionally, the expression of *TmSpz-like* mRNA was upregulated in the gut (Figure 4B) and fat bodies (Figure 4C) at 24 h post-infection with *E. coli*, *S. aureus*, and *C. albicans*. In the hemocytes, the expression of *TmSpz*-like mRNA was upregulated at 12 h post-infection with *C. albicans* (Figure 4D). The expression of *TmSpz*-like mRNA was upregulated in the Malpighian tubules at 24 h post-infection with *E. coli* and *S. aureus* (Figure 4E). These findings indicated that *TmSpz-like* mRNA is expressed in all the examined tissues and whole body of *T. molitor* after infection with various microorganisms.

### 2.5. TmSpz-like Knockdown Increased the Mortality of E. coli-Infected T. Molitor Larvae

To evaluate the role of *TmSpz-like* in the viability of *T. molitor* larvae, the ds*TmSpz-like*-treated larvae were infected with *E. coli*, *S. aureus*, and *C. albicans* (Figure 5). *T. molitor* larvae treated with enhanced green fluorescent protein (EGFP) dsRNA (ds*EGFP*) were used as the negative control. qRT-PCR analysis revealed that ds*TmSpz-like* but not ds*EGFP* significantly downregulated the expression of the *TmSpz-like* transcript in the *T. molitor* larvae. The expression of the *TmSpz-like* transcript in the ds*TmSpz-like*-injected larvae decreased by approximately 80% compared with the control group (ds*EGFP*) at day 4 post-injection (Figure 5A). The viability of *TmSpz-like*-silenced larvae was measured after infection with *E. coli*, *S. aureus*, and *C. albicans* (Figure 5B–D, respectively) for 10 days. *TmSpz-like-*knocked down larvae exposed to *E. coli* exhibited less than 60% viability (Figure 5B). The viability rates significantly decreased at day 8 post-*E. coli* infection. However, the viability rates of *S. aureus* and *C. albicans*-infected *TmSpz-like*-silenced larvae were similar to those of ds*EGFP*-injected larvae (Figure 5C,D, respectively). This suggested that the knockdown of *TmSpz-like* increased the mortality of *E. coli*-infected *T. molitor* larvae. Thus, *E. coli* infection affected the viability of the *TmSpz* knockdown larvae but not *S. aureus* or *C. albicans*.

### 2.6. Effects of TmSpz-like Knockdown on the Expression of AMPs and NF-κB

The findings of this study indicated that *TmSpz-like* plays an important role in the defense response against *E. coli*. *TmSpz-like* regulates AMP expression through the Toll signaling pathway. Thus, the expression levels of 15 different AMP-encoding genes in the whole body of *TmSpz-like* knockdown *T. molitor* larvae were examined at 6 h post-infection with *E. coli*, *S. aureus*, and *C. albicans* (Figure 6). Of the 15 AMP genes examined, the following 11 genes were significantly downregulated in the *E. coli*-infected *TmSpz-like* knockdown *T. molitor* larvae: *TmTene1*, *TmTene2*, *TmTene4*, *TmDef*, *TmDef-like*, *TmColeA*, *TmColeB*, *TmColeC*, *TmAtt1a*, *TmAtt1b*, and *TmAtt2*. The expression levels of *TmTene3*, *TmCec2*, *TmTLP1*, and *TmTLP2* were non-significantly different between the *E. coli*-infected and control *TmSpz-like* knockdown *T. molitor* larvae. Meanwhile, the expression levels of the following 13 AMP-encoding mRNAs were downregulated in the *S. aureus*-infected *TmSpz-like* knockdown *T. molitor* larvae: *TmTene1*, *TmTene2*, *TmTene4*, *TmDef*, *TmDef-like*, *TmCec2*, *TmColeA*, *TmColeB*, *TmColeC*, *TmAtt1a*, *TmAtt1b*, *TmAtt2*, and *TmTLP1*. The mRNA levels of *TmTene1*, *TmTene2*, *TmTene4*, *TmDef*, *TmDef-like*, *TmCec2*, *TmColeA*, *TmColeB*, *TmColeC*, *TmAtt1a*, *TmAtt1b*, *TmAtt2, TmTLP1*, and *TmTLP2* were slightly downregulated in the *C. albicans*-infected *TmSpz-like* knockdown *T. molitor* larvae (Figure 6).

To narrow down the effect of *TmSpz-like* on AMP expression, the expression levels of Toll and IMD NF-κB-related factors (*TmDorX1*, *TmDorX2*, and *TmRelish*) were measured under the same conditions (Figure 7). The mRNA expression levels of *TmDorX1*, *TmDorX2*, and *TmRelish* were downregulated in the *E. coli*-infected *TmSpz-like* knockdown *T. molitor* larvae. Similarly, the mRNA expression levels of *TmDorX1* and *TmDorX2* were downregulated in the *S. aureus*-infected and *C. albicans*-infected *TmSpz-like* knockdown *T. molitor* larvae. Furthermore, the mRNA levels of *TmRelish* were downregulated in the *E. coli*-infected *TmSpz-like* knockdown *T. molitor* larvae but not in the *S. aureus*-infected and *C. albicans*-infected *TmSpz-like* knockdown *T. molitor* larvae.

These findings indicate that *TmSpz-like* regulates the expression of 11 AMPs in response to *E. coli* infection in *T. molitor* larvae. The roles of NF-κB-related transcription factors such as *Dorsal* isoforms or *Relish* in the immune response to pathogen infection are unclear (Figure 8).

## 3. Discussion

The Toll receptor ligand Spz belongs to the family of proteins involved in the signaling pathway required for dorso-ventral patterning in the early embryo and antifungal response in the larvae and adults of *Drosophila* [24]. The homologs of *Spz* encode proteins with cysteine knot domains. The *Spz* gene family was derived from ancient gene duplication events, and the gene products activate the Toll receptor ligands. In *Drosophila*, Spz-4 and Spz-6, which are expressed in the late embryo, are involved in the antifungal response along with Toll 5 and development, respectively [25,26]. In *T. molitor*, Spz-4 is involved in the transcriptional regulation of AMPs, which modulate the humoral immune response against *E. coli* and *C. albicans* [9], whereas *T. molitor* Spz-6 protects the larvae against *E. coli* and *S. aureus* by positively regulating the expression of the AMPs Tenecin-2 and Tenecin-3 [11]. In this study, the full-length ORF sequence of *TmSpz-like* was identified and characterized using a molecular informatics approach. The cysteine motif domain of *Tm*Spz-like comprises conserved cysteine residues involved in disulfide bond formation, while one cysteine was involved in dimerization. A cysteine knot domain at the C-terminus, a cleavage region, and a signal peptide region has been identified in *Tm*Spz-4 and *Tm*Spz-6 [9,11]. Spz isoforms identified from the crustacean *L. vannamei* contain the signal peptide region and a cysteine knot region for Toll binding [18]. Serine protease mediates the cleavage of pro-Spz to generate activated Spz. The Spz cysteine-rich domains, which bind to the concave surface of the Toll leucine-rich repeat region, are involved in intracellular signal transduction. The activation of the Toll receptor by Spz mediates nuclear localization of the NF-κB/Rel transcription factor Dorsal, which promotes the transcription of AMPs and development-associated differentiation genes [27,28]. To examine the evolutionary relationships of *Tm*Spz-like with its homologs in other insects, phylogenetic analysis of the *Tm*Spz-like sequence was performed. *Tm*Spz-like exhibited the highest amino acid sequence similarity with the coleopteran *Tc*Spz, followed by the Hemipteran *Bi*Spz, *Ar*Spz-like, and *Nl*Spz. Previous studies have reported that *Tm*Spz-4 and *Tm*Spz-6 exhibited the least sequence similarity with *Tc*Spz [9,11].

In contrast to those of *TmSpz-4* and *TmSpz-6*, mRNA levels of *TmSpz-like* were upregulated in the egg, early larvae, and adult stages of *T. molitor* development. Additionally, the tissue distribution of *TmSpz-like* in the larval (maximum in the Malpighian tubules) and adult tissues (maximum in the gut) was different from that of *TmSpz-4* and *TmSpz-6.* The expression of *Spz* is regulated by hormonal changes during development, and pupal stages are associated with enhanced hormonal changes [29]. Hence, *TmSpz-like* expression is negatively correlated with hormonal changes. The upregulated expression of the *TmSpz-like* transcript during the egg stage may also be related to its functions, especially ventral-specific differentiation, during embryonic development, whereas the upregulated expression of *TmSpz-like* mRNA in the Malpighian tubules may be because the tissue does not undergo ecdysone-induced destruction as other larval tissues. In a non-infection model, AMPs rapidly respond to ecdysone in the tissue [30,31]. Furthermore, the expression of the Spz homolog was upregulated in the gut and fat body of the red palm weevil *R. ferrugineus* after systemic and oral inoculation of *S. aureus*, *E. coli*, and *Beauveria bassiana*, which suggested the role of the homolog Spz in systemic and gut-specific immune responses against infections [8]. *TmSpz-4* mRNA expression peaked at 9 and 24 h post-injection with *E. coli* and *S. aureus*, which confirmed the gut-specific immunity [9]. In this study, *TmSpz-like* mRNA exhibited the highest expression in the gut at 24 h post-inoculation with *E. coli* and *C. albicans*. *E. coli* was the only pathogen to promote the expression of *TmSpz-like* in the whole body of *T. molitor* larvae. The expression of *TmSpz-like* marginally increased in other tissues of *T. molitor*, which indicated that *E. coli* promoted Spz expression in *T. molitor*. Similar to *TmSpz-4* and *TmSpz-*6, *TmSpz-like* gene expression was upregulated after *E. coli* exposure [9,11]. Thus, *E. coli* (Gram-negative bacteria) can activate Spz and promote its binding to the Toll receptor in *T. molitor* and subsequently modulate the NF-κB-mediated transcriptional regulation of AMP-encoding genes. The silencing of *TmSpz-like* transcripts increased the mortality in *T. molitor* larvae after *E. coli* infection but not after *S. aureus* or *C. albicans* infection. This suggests that *TmSpz-like* is required for protection against *E. coli* infection in the larvae. Nevertheless, the knockdown of *TmSpz-like* did not result in cross-silencing effects on *TmSpz-4* and *TmSpz-6*. The sensitivity of *T. molitor* larvae with downregulated expression of *TmSpz-like*, *TmSpz-4*, and *TmSpz-6* to *E. coli* infections may be related to cross-talk between the Toll and IMD pathways. *E. coli* induces the activation of Spz in *T. molitor* larvae, suggesting that polymeric DAP-type PGN forms a complex with *Tenebrio* PGRP-SA to activate the Toll receptor ligand Spz [4,32]. The induction of the Toll pathway in *Drosophila* and *T. molitor* depends on the binding of Spz to Toll. The crustacean Toll directly binds to Gram-positive and Gram-negative bacteria and promotes Dorsal-mediated AMP expression [33]. In *M. sexta*, Gram-negative and Gram-positive bacteria activate a clip-domain proteinase proHP6 in the plasma. Subsequently, proHP6 activates proHP8, which activates Spz1 to bind to Toll and mediate the broad response to infection [34].

The knockdown of *TmSpz-like* significantly suppressed the expression of *TmTene1*, *TmTene2*, *TmTene4*, *TmDef*, *TmDef-like*, *TmColeA*, *TmColeB*, *TmColeC*, *TmAtt1a*, *TmAtt1b*, and *TmAtt2* in the *E. coli*-infected *T. molitor* larvae. The viability of *E. coli*-infected *TmSpz-like* knockdown *T. molitor* larvae may have decreased due to the downregulation of the AMP-encoding genes. These AMPs are reported to exert growth-inhibitory activity against *E. coli* [35,36,37,38,39,40]. *Tm*Tenecin3, an antifungal peptide, and *Tm*Thaumatin-like proteins are majorly induced in plant systems due to stress and other pathogenic insults [41,42]. These peptides were not downregulated in *TmSpz-like* knockdown larvae after infection with *E. coli*. The expression levels of *TmTene1*, *TmTene2*, *TmTene4*, *TmDef2*, *TmCole1*, *TmCole2*, *TmAtt1a*, *TmAtt1b*, and *TmAtta2* were downregulated in the *TmIMD* knockdown larvae after *E. coli* infection [22]. This indicates that the same AMPs are regulated under IMD and Toll pathways after *E. coli* infection. In the mosquito *A. aegypti* Aag2 cells, the AMP Gambicin is regulated by the combination of Toll, IMD, and JAK-STAT pathways [43]. To examine the regulation of AMP production and delineate the roles of NF-κB-mediated regulation of these AMPs, the expression of Dorsal isoforms (X1 and X2) and Relish in *TmSpz-like* knockdown larvae was examined after pathogenic challenges. The mRNA expression levels of *TmDorX1*, *TmDorX2*, and *TmRelish* were downregulated in *E. coli*-infected *TmSpz-like* knockdown larvae, which indicated that all NF-κB factors may regulate the 11 AMPs [44]. We speculated that *TmSpz-like* promotes the viability of *E. coli*-infected *T. molitor* by regulating the expression of 11 AMP-encoding genes. In addition to the Toll pathway-mediated NF-κB factors, *Tm*DorX1 and X2 and IMD-pathway-mediated *Tm*Relish may be involved in promoting viability. *Tm*Relish-mediated regulation of the 11 AMPs through the Toll pathway activated by *Tm*Spz-like must be elucidated.

## 4. Conclusions

In this study, we functionally characterized the *TmSpz-like* homolog from *T. molitor*. Previously, we had characterized the antifungal and antibacterial activities of *TmSpz-4* and *TmSpz-6*. *Tm*Spz-like exhibited the typical cysteine knot region with conserved cysteine residues. The expression of *TmSpz-like* mRNA was upregulated in the egg, early larvae, and adults. This was in contrast to the upregulated expression of *TmSpz-4* and *TmSpz-6* in the pupal stage. *E. coli* significantly decreased the viability of *TmSpz-like* knockdown larvae, which can be attributed to the downregulation of 11 AMP genes. Furthermore, the transcription factors involved in the Toll (*TmDorX1* and *X2*) and the IMD (*TmRelish*) pathways were downregulated in *E. coli*-infected *TmSpz-like* knockdown larvae.

## 5. Materials and Methods

### 5.1. Insect Rearing

Mealworm beetles (*T. molitor*) were reared at 27 ± 1 °C and 60% ± 5% relative humidity in an environmental chamber under dark conditions. The reared larvae were fed with an artificial diet containing 170 g wheat flour, 20 g roasted soy flour, 10 g protein, 100 g wheat bran, 0.5 g sorbic acid, 0.5 mL propionic acid, and 0.5 g chloramphenicol in 200 mL of distilled water. Healthy tenth to twelfth instar larvae were fed with the dietary mixture, which was pre-autoclaved at 121 °C for 15 min.

### 5.2. Microorganisms

The Gram-negative bacterium *Escherichia coli* K12, the Gram-positive bacterium *Staphylococcus aureus* RN4220, and the fungus *Candida albicans* AUMC 13529 were used for the immune challenge experiments. *E. coli* and *S. aureus* were cultured overnight in Luria-Bertani broth at 37 °C, while *C. albicans* was cultured in Sabouraud dextrose broth. The microorganisms were harvested, washed twice with phosphate-buffered saline (PBS; pH 7.0), and centrifuged at 5000 g for 15 min. Next, the samples were suspended in PBS, and the optical density of the suspension at 600 nm was measured using a spectrophotometer (Eppendorf, Germany). The density of *E. coli* and *S. aureus* was adjusted to 1 × 10^6^ cells/µL, while that of *C. albicans* was adjusted to 5 × 10^4^ cells/µL for immune challenge experiments.

### 5.3. In Silico Identification and Cloning of the Full-Length TmSpz-like cDNA

The *TmSpz-like* sequence was identified from the *T. molitor* RNAseq (unpublished) and expressed sequence tag databases. Local-tblastn analysis was performed using the amino acid sequence of *T. castaneum* Spz (XP_008201191.1) as a query. The deduced amino acid sequence of *Tm*Spz was analyzed using the BLASTx and BLASTp algorithms at the National Center for Biotechnology Information (https://blast.ncbi.nlm.nih.gov/Blast.cgi) with nr database. The full-length target open reading frame (ORF) regions were amplified using an AccuPower Pfu PreMix (Bioneer, Daejeon, South Korea) and a MyGenie 96 thermal block (Bioneer, Daejeon, South Korea) with gene-specific primers designed using Primer 3.0 software (http://bioinfo.ut.ee/primer3-0.4.0/) (Table 1). The PCR-purified products were cloned into the T-blunt vector cloning system (Solgent Company, Daejeon, Korea). The recombinant vector was transformed into *E. coli* DH5α cells and sequenced using M13 primers. The full-length ORF sequence was validated after sequencing.

### 5.4. Domain and Phylogenetic Analyses

The domain architecture of the protein sequences was retrieved using the InterProScan domain analysis (https://www.ebi.ac.uk/interpro/search/sequence-search) and BLASTp programs. The signal peptide was predicted using the SignalP 5.0 server (http://www.cbs.dtu.dk/services/SignalP/). Additionally, the three-dimensional (3D) structure of *Tm*Spz-like was predicted using the SWISS-MODEL server (https://swissmodel.expasy.org).

The ClustalX v. 2.1 [45]-based multiple sequence alignment profile was used to analyze the genetic relatedness among Spz-like sequences of different insect orders. The amino acid sequence of the cystine knot cytokine domain of *Tm*Spz-like was used for the analysis. The .pim output files from ClustalX v. 2.1 were used to analyze the percentage sequence similarity of Spz-like among orthologous species. A phylogenetic tree was constructed based on the amino acid sequences of *Tm*Spz-like using the neighbor-joining method with MEGA v. 7.0 [46]. The bootstrap consensus tree was inferred from 1000 replicates. The evolutionary distances were computed using the Poisson correction method. The amino acid sequence of *Penaeus vannamei* Spz5-like (*Pv*Spz5-like; XP_027217999) was used as an outgroup for this analysis.

### 5.5. Analysis of TmSpz-like Expression in Different Developmental Stages and Tissues

Total RNA was isolated from different developmental stages (eggs, young instar larvae (tenth to twelfth instar larvae), late instar larvae (nineteenth to twentieth instar larvae), pre-pupae, 1–7-day-old pupae, and 1–5-day-old adults), as well as from different tissues (integument, hemocytes, gut, fat body, and Malpighian tubules dissected from both late instar larvae and 5-day-old adults; ovary and testis from 5-day-old adults) of *T. molitor* using the Clear-S total RNA extraction kit (Invirustech Co., Gwangju, South Korea), following the manufacturer’s instructions. The isolated RNA (2 μg) was reverse transcribed into complementary DNA (cDNA) using Oligo(dT)_12-18_ primer. The reverse transcription reactions were performed using a MyGenie96 Thermal Block (Bioneer, Korea) and AccuPower^®^ RT PreMix (Bioneer, Korea), following the manufacturer’s instructions under the following conditions: 72 °C for 5 min, 42 °C for 1 h, and 94 °C for 5 min. cDNA was stored at −20 °C until analysis.

To investigate the effect of microbial infection on the expression of *TmSpz-like*, the twelfth to fifteenth instar larvae (n = 20) of *T. molitor* were infected with 1 × 10^6^ cells/µL of *E. coli* and *S. aureus* and 5 × 10^4^ cells/μL of *C. albicans*. The larvae belonging to the mock control group were injected with PBS. The whole larvae, hemocytes, gut, Malpighian tubules, and fat body were collected at 3, 6, 9, 12, and 24 h post-microbial inoculation. The relative expression level of *TmSpz-like* mRNA was investigated using qRT-PCR with AccuPower^®^ 2X Greenstar^TM^ qPCR Master Mix (Bioneer, Korea), synthesized cDNAs as templates, and *TmSpz-like*-specific primers designed using the Primer 3 plus program (http://primer3plus.com/cgi-bin/dev/primer3plus.cgi) (*Tm*Spz_qPCR_Fw and *Tm*Spz qPCR_Rv) (Table 1). The PCR conditions were as follows: initial denaturation at 95 °C for 5 min, followed by 40 cycles of 95 °C for 15 s (denaturation) and 60 °C for 30 s (annealing and extension). qRT-PCR analysis was performed using an AriaMx Real-Time PCR System (Agilent Technologies, Santa Clara, CA, USA), and the data were analyzed using AriaMx Real-Time PCR software. *T. molitor* ribosomal protein L27a-encoding gene (*TmL27a*) was used as an internal control. The mRNA expression levels were analyzed using the 2^−∆∆Ct^ method [47]. The results are represented as mean ± standard error from three biological replications.

### 5.6. RNAi Analysis

To synthesize the double-stranded (ds) RNA fragment of *TmSpz-like*, the *TmSpz-like* DNA fragment was PCR-amplified using gene-specific primers with a T7 promoter sequence at the 5′ end (Table 1). The primers were designed using the SnapDragon software (http://www.flyrnai.org/cgi-bin/RNAi_find_primers.pl) to prevent any cross-silencing effects. The fragments were amplified using AccuPower^®^ Pfu PCR PreMix under the following conditions: initial denaturation step at 94 °C for 5 min; followed by 35 cycles at 94 °C for 30 s (denaturation), 53 °C for 30 s (annealing), and 72 °C for 30 s (extension); and a final extension at 72 °C for 5 min. The PCR products were purified using the AccuPrep PCR purification kit (Bioneer, Daejeon, Korea), while dsRNA was synthesized using the EZ^TM^ T7 high yield in vitro transcription kit (Enzynomics, Daejeon, Korea), following the manufacturer’s instructions. The dsRNA product was purified through precipitation with 5 M ammonium acetate and 99% ethanol and quantified using an Epoch spectrophotometer (BioTek Instruments, Inc., VT, USA). The synthesized dsRNA was stored at −20 °C until analysis.

To validate the knockdown of *TmSpz-like*, 1 µg/µL of synthesized ds*EGFP* was injected into young instar larvae (tenth to twelfth instars; n = 30) using disposable capillary needles mounted on a micro-applicator (Picospiritzer III Micro Dispense System; Parker Hannifin, NH, USA). ds*EGFP* was used as a negative control.

### 5.7. Effect of TmSpz-like Knockdown on Transcriptional Regulation of AMP and NF-κB

To further characterize the function of *TmSpz-like* in innate immunity, the effects of *TmSpz-like* knockdown on the expression levels of 15 *T. molitor* AMP-encoding and three NF-κB transcription factor-encoding genes were examined after microbial challenge. *TmSpz-like* expression in larvae was knocked down using the RNAi technique. The *TmSpz-like* knockdown larvae were infected with *E. coli*, *S. aureus*, and *C. albicans*. The ds*EGFP*- and PBS-treated larvae served as the negative and injection controls, respectively. The larvae were homogenized at 6 h post-injection. Total RNA was extracted from the homogenized samples and subjected to cDNA synthesis as described above. The expression levels of 15 AMP genes (*TmTenecin-1* (*TmTene-1*), *TmTenecin-2* (*TmTene-2*), *TmTenecin-3* (*TmTene-3*), *TmTenecin-4* (*TmTene-4*), *TmAttacin-1a* (*TmAtt-1a*), *TmAttacin-1b* (*TmAtt-1b*), *TmAttacin-2* (*TmAtt-2*), *TmDefensin* (*TmDef*), *TmDefensin-like* (*TmDef-like*), *TmColeoptericin-A* (*TmCole-A*), *TmColeoptericin-B* (*TmCole-B*), *TmColeoptericin-C* (*TmCole-C*), *TmCecropin-2* (*TmCec-2*), *TmThaumatin-like protein-1* (*TmTLP-1*), and *TmThaumatin-like protein-2* (*TmTLP-2*)) and three NF-κB factors involved in the Toll and IMD pathways (*TmDorsal isoform X1* (*TmDorX1*), *TmDorsal isoform X2* (*TmDorX2*), and *TmRelish* (*TmRel*)) were examined. The qRT-PCR analysis was performed using *TmDorsal* and *TmRelish*-specific primers (Table 1). All experiments were performed in triplicate.

### 5.8. Statistical Analysis

All experiments were performed in triplicate. The data were analyzed using one-way analysis of variance with SAS 9.4 software (SAS Institute, Inc., NC, USA). Cumulative viable rates were compared using Tukey’s multiple range test. Differences were considered significant at *p* < 0.05. The expression levels of the target genes were normalized with those of the internal control (*TmL27a*) and external control (PBS). The relative expression levels were calculated using the 2 ^−(∆∆Ct)^ method.

## Figures and Tables

**Figure 1 ijms-22-10888-f001:**
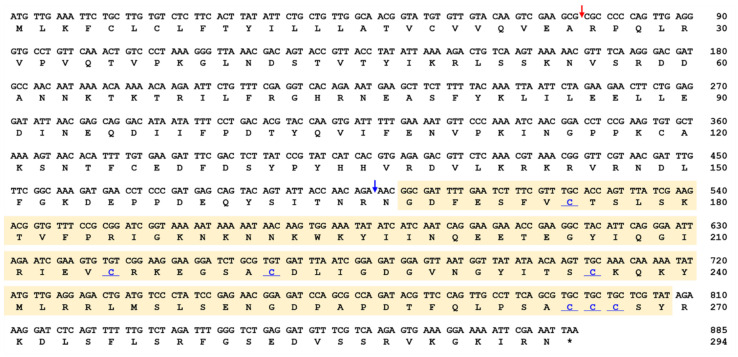
Nucleotide and deduced amino acid sequences of *TmSpz-like*. *TmSpz-like* contains an 885 bp open reading frame and encodes a predicted polypeptide of 294 amino acid residues. Domain analysis revealed that *Tm*Spz-like includes a cysteine knot domain (yellow shaded box), a signal peptide region cleaved after 25 amino acid residues (red arrow), and a cleavage site (blue arrow). The conserved cysteine residues of the cysteine knot domain forming three disulfide bridges are shown in blue font. Numbers on the right of each row represent nucleotide and amino acid positions.

**Figure 2 ijms-22-10888-f002:**
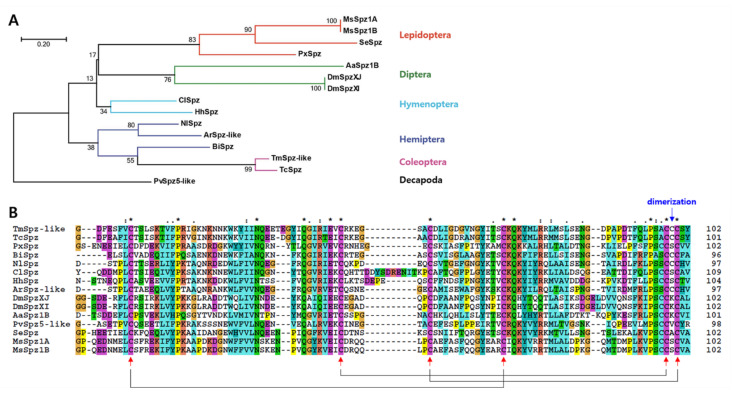
Molecular phylogenetic analysis of *Tm*Spz-like and its homologs in other insects. (**A**) Maximum likelihood-based phylogenetic tree based on the amino acid sequence of *Tm*Spz-like constructed using the MEGA 7.0 software. The numbers above the branches represent bootstrap values (1000 replications). (**B**) Multiple alignments of the deduced amino acid sequence of *Tm*Spz-like aligned using Clustal X 2.0.11. Identical and similar sites are identified with asterisks (*) and dots (: or .), respectively. Arrows represent the phylogenetically conserved cysteine residues forming the disulfide bridges featuring in the cysteine knot domain. The following protein sequences were used in the multiple sequence alignment and phylogenetic analysis: *Tc*Spz, *Tribolium castaneum* Spätzle (XP_008201191. 1); *Se*Spz, *Spodoptera exigua* Spätzle (KAF9408664. 1); *Px*Spz, *Plutella xylostella* Spätzle isoform X1 (XP_037962889. 1); *Ms*Spz1A, *Manduca sexta* Spätzle 1A (ACU68553. 1); *Ms*Spz1B, *M. sexta* Spätzle 1B (ACU68554. 1); *Nl*Spz, *Neodiprion lecontei* Spätzle (XP_015518347. 1); *Ar*Spz-like, *Athalia rosae* Spätzle-like (XP_012257981. 1); *Bi*Spz, *Bombus impatiens* Spätzle (XP_024228470. 1); *Cl*Spz, *Cimex lectularius* Spätzle isoform X2 (XP_024082501. 1); *Hh*Spz, *Halyomorpha halys* Spätzle (KAE8573434. 1); *Aa*Spz1B, *Aedes aegypti* Spätzle 1B (NP_001350875. 1); *Dm*SpzXJ, *Drosophila melanogaster* Spätzle isoform J (NP_733192. 1); *Dm*SpzXI, *D. melanogaster* Spätzle isoform I (NP_733188. 1); *Pv*Spz5-like, *Penaeus vannamei* Spz5-like (XP_027217999).

**Figure 3 ijms-22-10888-f003:**
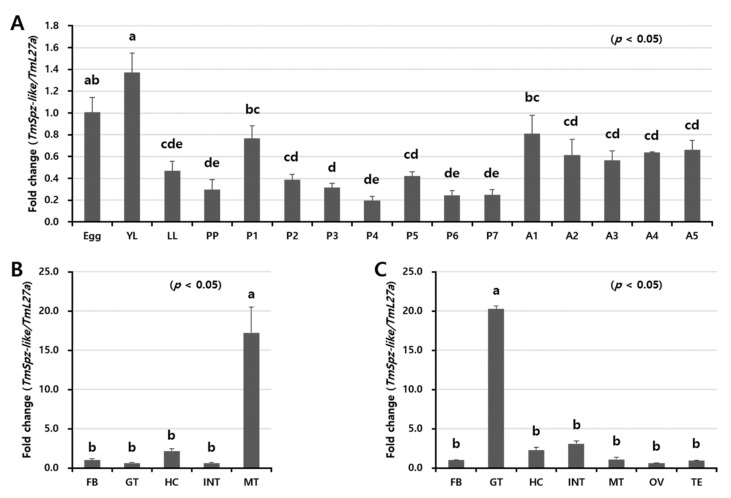
Relative expression levels of *TmSpz-like* mRNA in different developmental stages and tissues of *Tenebrio molitor*. (**A**) Expression levels of *TmSpz-like* in *T. molitor* at the egg, the young larval (YL), late larval (LL), pre-pupal (PP), 1–7-day-old pupal (P1–7), and 1–5-day-old adult (A1–5) stages. Tissue distribution of *TmSpz-like* transcripts in late larvae (**B**) and five-day-old adults (**C**) analyzed using quantitative real-time polymerase chain reaction. *T. molitor* 60S ribosomal protein L27a (*TmL27a*)-encoding gene was used as an internal control. Fat body (FB), gut (GT), hemocytes (HC), integument (INT), and Malpighian tubules (MT) of late instar larvae and adults, in addition to ovaries (OV) and testes (TE) of adults, were dissected and collected from 20 late larvae and 5-day-old adults for analysis. Vertical bars represent mean ± standard error from three biological replicates. Data were analyzed using one-way analysis of variance, followed by Tukey’s multiple range tests at 95% confidence level (*p* < 0.05). The graphs indicated by the same letter (a, b, bc, c, cd, d, de, e) are not significantly different (Tukey’s multiple range; *p* < 0.05).

**Figure 4 ijms-22-10888-f004:**
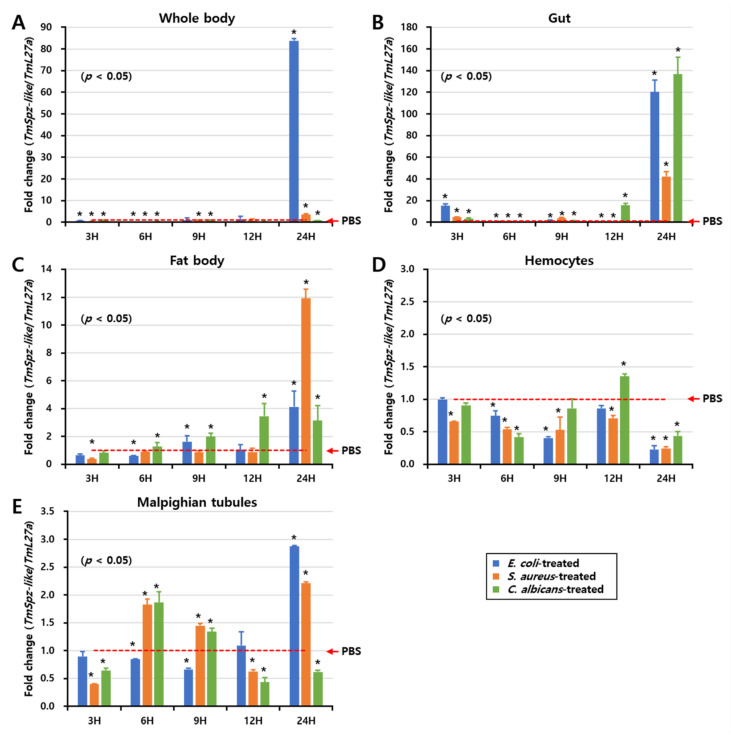
*TmSpz-like* mRNA expression profiles after microbial challenge. *TmSpz-like* relative expression levels in the whole body (**A**), gut (**B**), fat body (**C**), hemocytes (**D**), and Malpighian tubules (**E**) of larvae infected with *Escherichia coli*, *Staphylococcus aureus*, and *Candida albicans.* The mock control group comprised phosphate-buffered saline-treated larvae. Expression levels of *TmSpz-like* mRNA in the mock control group were normalized to 1. Vertical bars represent mean ± standard error from three biological replicates. Significant differences between the experimental and control groups are indicated by asterisks (* *p* < 0.05).

**Figure 5 ijms-22-10888-f005:**
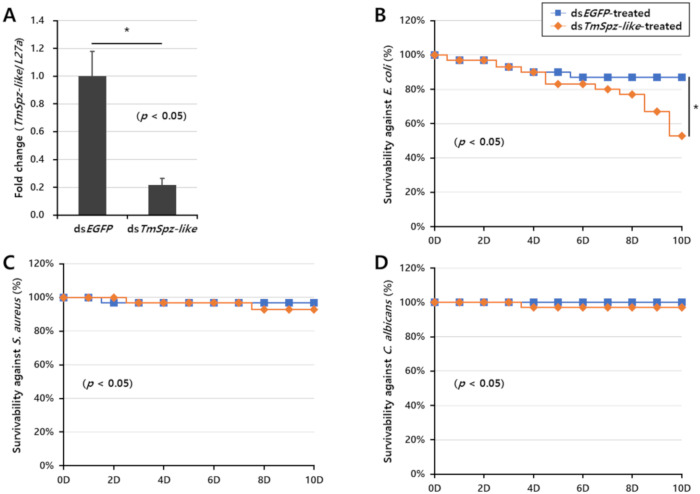
Effect of *TmSpz-like* knockdown on the viability of *Tenebrio molitor* larvae. (**A**) *TmSpz-like* knockdown efficiency measured using quantitative real-time polymerase chain reaction at day 4 post-injection. Viability of *TmSpz-like* knockdown larvae after challenge with *Escherichia coli* (**B**), *Staphylococcus aureus* (**C**), or *Candida albicans* (**D**) (n = 30). The negative control comprised ds*EGFP*-injected larvae. Data are presented as average of three biologically independent replicate experiments. Asterisks indicate significant differences between *TmSpz-like*-knocked down and ds*EGFP-*treated groups (*p* < 0.05). Survival rates were analyzed based on the Kaplan–Meier plots (log-rank chi-square test; * *p* < 0.05).

**Figure 6 ijms-22-10888-f006:**
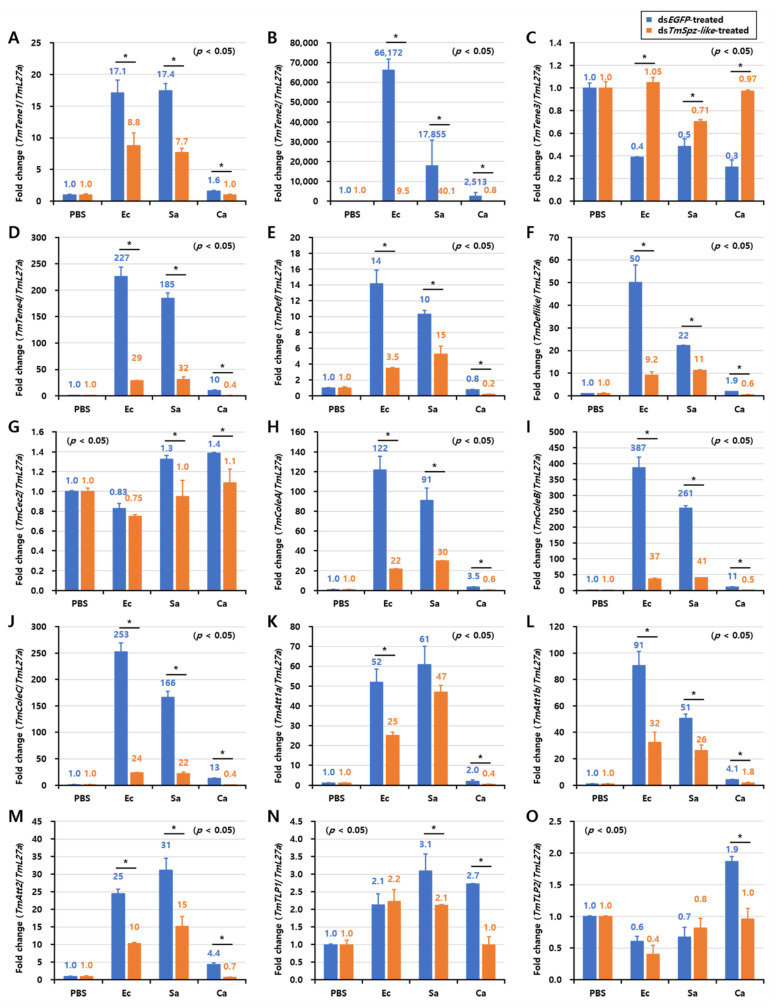
Antimicrobial peptide (AMP)-encoding mRNA expression patterns in whole body of *TmSpz-like* knockdown larvae in response to *Escherichia coli, Staphylococcus aureus*, and *Candida albicans* infections. Phosphate-buffered saline (PBS) was injected as a control at day 4 post-*TmSpz-like* knockdown. At 6 h post-microbial challenge, the expression levels of AMP-encoding genes, including *TmTene1* (**A**), *TmTene2* (*B*), *TmTene3* (**C**), *TmTene4* (**D**), *TmDef* (**E**), *TmDef-like* (**F**), *TmCec2* (**G**), *TmColeA* (*H*), *TmColeB* (**I**), *TmColeC* (**J**), *TmAtt1a* (**K**), *TmAtt1b* (**L**), *TmAtt2* (**M**), *TmTLP1* (**N**), and *TmTLP2* (**O**), were measured using quantitative real-time polymerase chain reaction. ds*EGFP* was injected as a negative control. *TmL27a* expression was measured as an internal control. The number above the bars represents the AMP transcriptional levels. The experiments were performed in biological triplicates. Asterisks indicate significant differences between ds*TmSpz-like*- and ds*EGFP*-injected groups (Student’s *t*-test; * *p* < 0.05).

**Figure 7 ijms-22-10888-f007:**
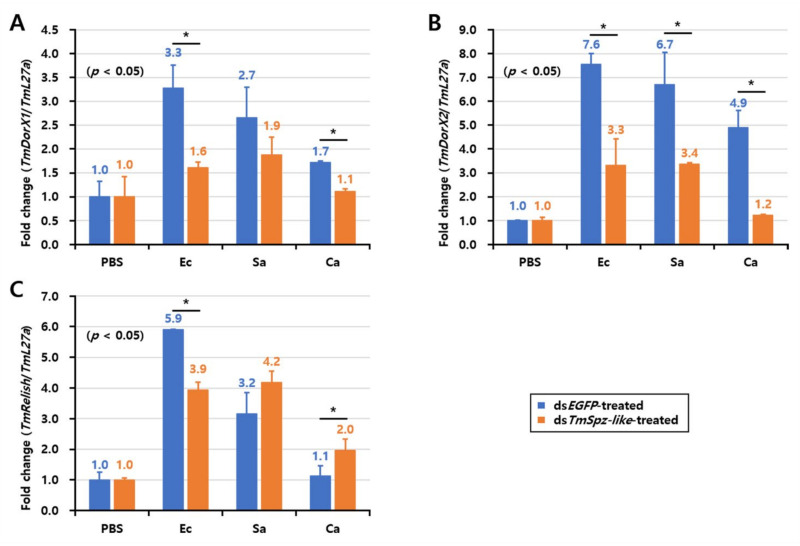
Effect of *TmSpz-like* knockdown on the expression of nuclear factor kappa B (NF-κB)-encoding genes after challenge with *Escherichia coli, Staphylococcus aureus*, and *Candida albicans*. mRNA expression levels of *TmDorX1* (**A**), *TmDorX2* (**B**), and *TmRelish* (**C**) were investigated using quantitative real-time polymerase chain reaction. The negative control comprised ds*EGFP*-injected larvae. *TmL27a* expression was assessed as an internal control. All experiments were performed in triplicate. Asterisks indicate significant differences in NF-κB-encoding gene expression between the ds*TmSpz-like* and ds*EGFP-*injected groups (Student’s *t*-test; * *p* < 0.05).

**Figure 8 ijms-22-10888-f008:**
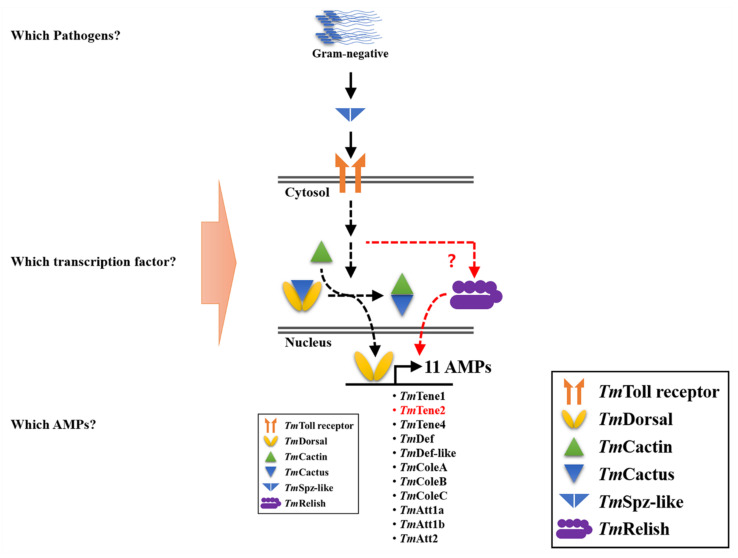
An illustrative summary of the humoral immune pathway positively regulated by *TmSpz-like* in *Escherichia coli*-infected *Tenebrio molitor.* Eleven antimicrobial peptide (AMP)-encoding genes were regulated by *TmSpz-like*, which indicated that *TmSpz-like* is required for the survival of the *E. coli*-infected host.

**Table 1 ijms-22-10888-t001:** Primers used in this study.

Name	Primer Sequences (5′-3′)
*Tm*Spz-like-ORF-Fw	CGTTTTCAGCGGCTAATTGT
*Tm*Spz-like-ORF-Rv	CATAATTCCCTTTTCCCAATTT
*Tm*Spz-like-T7-Fw	TAATACGACTCACTATAGGGT ATGTTCCCAAAATCAACGGA
*Tm*Spz-like-T7-Rv	TAATACGACTCACTATAGGGT AATCACACGCAGATCCTTCC
EGFP-T7-Fw	TAATACGACTCACTATAGGGT CGTAAACGGCCACAAGTTC
EGFP-T7-Rv	TAATACGACTCACTATAGGGT TGCTCAGGTAGTGTTGTCG
*Tm*Spz-like-qPCR-Fw	CAGTTGAGGGTGCCTGTTCA
*Tm*Spz-like-qPCR-Rv	TTGTTGGCATCGTCCCTTGA
*Tm*L27a-qPCR-Fw	TCATCCTGAAGGCAAAGCTCCAGT
*Tm*L27a-qPCR-Rv	AGGTTGGTTAGGCAGGCACCTTTA
*Tm*Tenecin-1-qPCR-Fw	CAGCTGAAGAAATCGAACAAGG
*Tm*Tenecin-1-qPCR-Rv	CAGACCCTCTTTCCGTTACAGT
*Tm*Tenecin-2-qPCR-Fw	CAGCAAAACGGAGGATGGTC
*Tm*Tenecin-2-qPCR-Rv	CGTTGAAATCGTGATCTTGTCC
*Tm*Tenecin-3-qPCR-Fw	GATTTGCTTGATTCTGGTGGTC
*Tm*Tenecin-3-qPCR-Rv	CTGATGGCCTCCTAAATGTCC
*Tm*Tenecin-4-qPCR-Fw	GGACATTGAAGATCCAGGAAAG
*Tm*Tenecin-4-qPCR-Rv	CGGTGTTCCTTATGTAGAGCTG
*Tm*Defensin-Fw	AAATCGAACAAGGCCAACAC
*Tm*Defensin-Rv	GCAAATGCAGACCCTCTTTC
*Tm*Defensin-like-Fw	GCGATGCCTCATGAAGATGTAG
*Tm*Defensin-like-Rv	CCAATGCAAACACATTCGTC
*Tm*Coleoptericin-A-qPCR-Fw	GGACAGAATGGTGGATGGTC
*Tm*Coleoptericin-A-qPCR-Rv	CTCCAACATTCCAGGTAGGC
*Tm*Coleoptericin-B-qPCR-Fw	CAGCTGTTGCCCACAAGTG
*Tm*Coleoptericin-B-qPCR-Rv	CTCAACGTTGGTCCTGGTGT
*Tm*Coleoptericin-C-qPCR-Fw	GGACGGTTCTGATCTTCTTGAT
*Tm*Coleoptericin-C-qPCR-Rv	CAGCTGTTTGTTTGTTCTCGTC
*Tm*Attacin-1a-Fw	GAAACGAAATGGAAGGTGGA
*Tm*Attacin-1a-Rv	TGCTTCGGCAGACAATACAG
*Tm*Attacin-1b-Fw	GAGCTGTGAATGCAGGACAA
*Tm*Attacin-1b-Rv	CCCTCTGATGAAACCTCCAA
*Tm*Attacin-2-Fw	AACTGGGATATTCGCACGTC
*Tm*Attacin-2-Rv	CCCTCCGAAATGTCTGTTGT
*Tm*Cecropin-2-Fw	TACTAGCAGCGCCAAAACCT
*Tm*Cecropin-2-Rv	CTGGAACATTAGGCGGAGAA
*Tm*Thaumatin-like protein-1-Fw	CTCAAAGGACACGCAGGACT
*Tm*Thaumatin-like protein-1-Rv	ACTTTGAGCTTCTCGGGACA
*Tm*Thaumatin-like protein-2-Fw	CCGTCTGGCTAGGAGTTCTG
*Tm*Thaumatin-like protein-2-Rv	ACTCCTCCAGCTCCGTTACA
*Tm*DorX1_qPCR_Fw	AGCGTTGAGGTTTCGGTATG
*Tm*DorX1_qPCR_Rv	TCTTTGGTGACGCAAGACAC
*Tm*DorX2_qPCR_Fw	ACACCCCCGAAATCACAAAC
*Tm*DorX2_qPCR_Rv	TTTCAGAGCGCCAGGTTTTG
*Tm*Relish_qPCR_Fw	AGCGTCAAGTTGGAGCAGAT
*Tm*Relish_qPCR_Rv	GTCCGGACCTCAAGTGT

Underline indicates T7 promotor sequences.

## Data Availability

Data is contained within the article or Appendix A.

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
