# Peer review of "TmSpz-like Plays a Fundamental Role in Response to E. coli but Not S. aureus or C. albican Infection in Tenebrio molitor via Regulation of Antimicrobial Peptide Production"

_ijms, 2021, doi:10.3390/ijms221910888_

Round 1

Reviewer 1 Report

In the manuscript entitled, “TmSpz-like regulates the production of antimicrobial peptides in Escherichia coli-infected Tenebrio molitor” Jang et al have characterized the role of TmSpZ-like protein of Tenebrio molitor in the immune response to E. coli, Staphylococcus aureus and C. albicans. The authors demonstrate that TmSpz-like is differentially expressed during the development of the insect and is induced upon infection. They further show that silencing the expression of TmSpz-like results in decreased survival of T. molitor possibly due to reduction in the release of antimicrobial peptides and the process is mediated through the NFkB pathway. While the results are extensive and well represented, the study presents some major weaknesses. While the authors have identified a new protein with properties similar to the Spätzle proteins, the manuscript lacks novelty. The role of multiple Spz proteins has been previously demonstrated and the manuscript apparently seems like a replica of the previous manuscripts published by the same group. In addition, the mechanism of action of Spz-like protein as demonstrated by the authors is not very clear and does not support the hypothesis of being specific to gram-negative bacteria. The entire manuscript is based on real-time PCR data and direct correlation to signaling cascades cannot be simply made using real-time PCR data. The authors must address the following concerns to improve the manuscript.

Major concerns:

  1. In figure 4, the expression of TmSpz-like is regulated upon infection with all the microorganisms that were utilized for infection. However, the authors in the subsequent figures focus on the specific effects of gram-negative bacteria. How do the authors justify this?
  2. In figure 5, what is the effect of infection by Staphylococcus aureus or Candida albicans beyond 10 days of infection? Why was day 10 selected for the survival assessment?
  3. It appears that Staphylococcus aureus increases the expression of many AMPs similar to E. coli infection which is abrogated by depletion of TmSpz-like. While the authors show in figure 5 that the infection by E. coli alters the survival of the insect and Staphylococcus aureus or Candida albicans had no effect, how do the authors justify the regulation of the expression of AMP-encoding mRNAs by E. coli and S. aureus infection as seen in figure 6?
  4. Why were the expression of AMPs and nuclear factor kappa B (NF-κB)-encoding genes assessed only in the whole body in figures 6 and 7 while regulation of TmSpz-like was observed in the gut and malpighian tubules? The authors must include the expression of AMPs and nuclear factor kappa B (NF-κB)-encoding genes in these tissues as well. Did the authors observe any difference in the expression of AMPs and nuclear factor kappa B (NF-κB)-encoding genes upon E. coli or S. aureus in these tissues?
  5. What is the similarity between the different TmSpzs (line 91) and TmSpz-like? How many domains and cysteine residues are conserved among them?
  6. Does knockdown of TmSpZ-like affect the expression of the other identified TmSpzs? Is there any redundancy in their expression pattern?
  7. For the most part of the data in the manuscript, the authors have assessed the mRNA levels instead of protein levels. It is advisable to check the levels of signaling proteins that participate in the NFkB signaling instead of their mRNA levels.
  8. The authors must include another gram-negative bacteria to demonstrate the robustness of the role of TmSpz-like in molitor.
  9. The number of insects used for the survival studies in figure 5 must be included as it is not clear from the legends.

Minor concerns:

  1. The title includes only E. coli while the manuscript includes Staphylococcus aureus or Candida albicans infection as well. The authors must modify the title accordingly.
  2. In the legend for figure 5b,c, and d, the color code is missing.
  3. The supplementary figure 1 is critical for annotation of the Spz-like protein and may be moved to the main figure.
  4. The authors must utilize asterisk/ns to denote the significance instead of the alphabets for the clarity of the readers.
  5. In figure 5A, what is the relative expression of TmSpz-like in un-injected molitor as compared to ds-EGFP injected insect larvae?

Author Response

Reviewer #1 (Comments to the Author): 

In the manuscript entitled, “TmSpz-like regulates the production of antimicrobial peptides in Escherichia coli-infected Tenebrio molitor” Jang et al have characterized the role of TmSpZ-like protein of Tenebrio molitor in the immune response to E. coli, Staphylococcus aureus and C. albicans. The authors demonstrate that TmSpz-like is differentially expressed during the development of the insect and is induced upon infection. They further show that silencing the expression of TmSpz-like results in decreased survival of T. molitor possibly due to reduction in the release of antimicrobial peptides and the process is mediated through the NFkB pathway. While the results are extensive and well represented, the study presents some major weaknesses. While the authors have identified a new protein with properties similar to the Spätzle proteins, the manuscript lacks novelty. The role of multiple Spz proteins has been previously demonstrated and the manuscript apparently seems like a replica of the previous manuscripts published by the same group. In addition, the mechanism of action of Spz-like protein as demonstrated by the authors is not very clear and does not support the hypothesis of being specific to gram-negative bacteria. The entire manuscript is based on real-time PCR data and direct correlation to signaling cascades cannot be simply made using real-time PCR data. The authors must address the following concerns to improve the manuscript.

Major concerns:

  1. In figure 4, the expression of TmSpz-like is regulated upon infection with all the microorganisms that were utilized for infection. However, the authors in the subsequent figures focus on the specific effects of gram-negative bacteria. How do the authors justify this?

Authors Response: We own the pleasure of receiving your constructive comments on this manuscript. Regarding your comment above, kindly be informed that within both relevant figure legend and result, TmSpz-like expression induced by all pathogens has been described. However, Most of the TmSpz-like expression in whole body and gut was post-challenge with E. coli (Gram-negative bacteria). RNAi gene silencing studies highlighted the survival of the larvae post-challenge with E. coli with significant changes in survivability. Therefore, we proceed further with our other experimental sections the focused of our conclusion has been narrowed down to Gram-negative bacteria of our interest E. coli.

  1. In figure 5, what is the effect of infection by Staphylococcus aureus or Candida albicans beyond 10 days of infection? Why was day 10 selected for the survival assessment?

  1. It appears that Staphylococcus aureus increases the expression of many AMPs similar to E. coli infection which is abrogated by depletion of TmSpz-like. While the authors show in figure 5 that the infection by E. coli alters the survival of the insect and Staphylococcus aureus or Candida albicans had no effect, how do the authors justify the regulation of the expression of AMP-encoding mRNAs by E. coli and S. aureus infection as seen in figure 6?

Authors Response: Thank you for this fine comment. Please consider that among all the tested AMPs, tenesins and defensins unraveled to have anti Gram-positive bacteria characteristics. Considering our results of AMP gene expression, it has been illustrated in figure 6, that TmTene3 not only did not show downregulation changes but also showed upregulation after S. aureus infection, proposing that once TmSpz-like is knocked down, due to some possible hemostasis activity, upregulation of other genes effect TmTene3 gene expression and these possible mechanisms lead to survival of insects against S. aureus infection.

  1. Why were the expression of AMPs and nuclear factor kappa B (NF-κB)-encoding genes assessed only in the whole body in figures 6 and 7 while regulation of TmSpz-like was observed in the gut and malpighian tubules? The authors must include the expression of AMPs and nuclear factor kappa B (NF-κB)-encoding genes in these tissues as well. Did the authors observe any difference in the expression of AMPs and nuclear factor kappa B (NF-κB)-encoding genes upon E. coli or S. aureus in these tissues?

Authors Response: Thank you so much for your fine critic. Please be informed that for preliminary characterization of TmSpz-like, one of our goals was to investigate the TmSpz-like function in response to systemic infection. Therefore, checking the relevancy of the gene of interest expression with AMP genes and NF-kB response element in whole body is sensible enough for basic identifications and hypothesis regarding the systemic infection. Having said that, having the highest expression observed in gut post-challenge with E. coli, it is worthy to be studied after oral infection. Accordingly, TmSpz-like role in gut immunity shall be investigated separately in the future studies.

  1. What is the similarity between the different TmSpzs (line 91) and TmSpz-like? How many domains and cysteine residues are conserved among them?

Authors Response: Kindly know that the amino acid sequences of cystine knot domain predicted from T. molitor Spz genes were analyzed by multiple sequence alignment using clustal X2 program. The sequence similarity of cystine knot domains were low (5-52%). Whereas the cysteine residues which are important in structure formation, were conserved. Please find the relevant data as follows:

  1. Does knockdown of TmSpZ-like affect the expression of the other identified TmSpzs? Is there any redundancy in their expression pattern?

Authors Response: We could not be more grateful for your effectual comments and notices. Considering your comment above, Kindly be informed that Spaetzle is a conserved protein within different spices and due to this conservation in Tenebrio molitor, the redundancy in other Spaetzle expression after knocking down one of them is inevitable. However, in this very case, please note that subsequent to TmSpz-like knock down, TmSpz5 and TmSpz6 expression showed significant fall as well. Nevertheless, post infection expression of TmSpz5 and TmSpz6 upregulated despite of the fact that TmSpz-like remained knocked down. Please find the relevant data as follows:

  1. For the most part of the data in the manuscript, the authors have assessed the mRNA levels instead of protein levels. It is advisable to check the levels of signaling proteins that participate in the NFkB signaling instead of their mRNA levels.

Authors Response: We deeply appreciate your sensible point in this matter. Given the importance of mRNA as an information carrying molecule, please mark this fact that mRNA expression results is great proof of our genetic code directly from DNA to ribosomes, protein-making machinery. However, we did not stop our investigation here, therefore, we have checked NF-kB-encoding genes, which act downstream of ligand Spaetzle and upstream of AMP genes. Kindly note that without activating NF-kB relevant signaling proteins by ligand Spaetzle, alteration in transcription response elements would not be observed. More crucially, in this precise matter, our NF-kB gene expression data were in accordance with AMP assay results and relevant conclusion was made accordingly. Albeit, your kind suggestions definitely shall be considered in our future directions and studies.

  1. The authors must include another gram-negative bacteria to demonstrate the robustness of the role of TmSpz-like in molitor.

Authors Response: We appreciate your concern regarding the referred matter. Kindly note that our experimental design has been made based on the purpose of preliminary characterization of T. molitor Spz-like. Ergo, result of each experiment conducts the subsequent one. Nevertheless, relations of TmSpz-like knock down and E. coli infection turned to be the major discovery of this study. Investigating other Gram-negative bacterial infection and function of ligand Spaetzles shall be carried out in different criteria. Considering your comment above, TmSpz-like function in response to diverse pathogenic source including but not limited to entomopathogens or non-entomopathogens Gram-negative bacteria will be a great aim for further studies we will pursue in the future.

  1. The number of insects used for the survival studies in figure 5 must be included as it is not clear from the legends.

Authors Response: Thank you for another fine notice. Please note that the referred issue has been taken care of according to your comment. Please find the relevant change in Line 232 as it is mentioned n=30.

Minor concerns:

  1. The title includes only E. coli while the manuscript includes Staphylococcus aureus or Candida albicans infection as well. The authors must modify the title accordingly.

Authors Response: Thank you for this fine comment. Please consider that the title of the manuscript has been changed according to your request as follows:

TmSpz-like play fundamental role in response to E. coli but not S. aureus or C. albican infection in Tenebrio molitor via regulation of antimicrobial peptides production.

  1. In the legend for figure 5b,c, and d, the color code is missing.

Authors Response: Thank you for your comment. Kindly notice that color code related to Figure 5 has been added according to your inquiries.

  1. The supplementary figure 1 is critical for annotation of the Spz-like protein and may be moved to the main figure.

Authors Response: Thank you for your comment. With all respect, kindly note that considering the main focus of our study and experimental objectives, the referred data set is the structural model of protein Spaetzle-like and it is more preferable to remain in supplementary data section.

  1. The authors must utilize asterisk/ns to denote the significance instead of the alphabets for the clarity of the readers.

Authors Response: We appreciate your concern regarding the above matter. Kindly note that as the analysis used for figure 3 data set is Tukey’s multiple range and multiple means are found to show their significant difference with each other. Therefore, different letters have been applied in order readers be able to comprehend the difference of multiple groups compared with each other simultaneously.

  1. In figure 5A, what is the relative expression of TmSpz-like in un-injected molitor as compared to ds-EGFP injected insect larvae?

Authors Response: We appreciate one more your time all your constructive comments. Regarding your last comment, please note that in figure 5A we have use dsEGFP as the control, in order to normalize the effects of double strand injection itself. Fig. 5A shows the RNAi gene silencing validation results.   

Reviewer 2 Report

In this study, the authors identify an ORF from T. molitor encoding a Spätzle-like protein. Initially, the authors sequence the gene and perform a bioinformatic characterization. Next, the authors analyze the expression of the peptide in the insect tissues and during its development. The authors focus their attention on the regulation of the peptide following infection with three pathogens (one gram-positive, one gram-negative and one yeast). Finally, using knockdown mutant for the TmSpz-like gene, the authors reconstruct the regulatory pathway associated with toll receptors and the innate immune response.

The manuscript is well-conceived, the results are solid and interesting. However, some points need to be clarified, especially regarding the bioinformatics approaches used. Moreover, there is some imprecision (for example, the legends of the figures must be reviewed and uniformly formatted). 

Comments

  • I have some doubts about the computational data indicated in the study. The computational model built with SwissModel appears to be of low quality (according to the colours used by the webserver). The sequence homology between the TmSpz-like protein and the homolog 3e07 is unclear. If it is too low, authors should use another modelling tool, such as ab-initio modelling or a tool that uses both approaches (ab-initio and homology). 
  • Line 111: What codon usage data was used to infer the amino acid sequence? There may be differences due to the different types of use of codons used as a reference.
  • Fig. 4 is significant, but there are some unclear details. For example, why did the authors choose these fabrics? Is there any evidence of a link between these tissues and immunity? Also, why is E. coli so high in the “whole body” sample and not so high in other tissues? what is a possible explanation for this?
  • The N-terminal signal sequences were predicted with SignalP: include the results in Supplementary materials. Also, the domain organization was predicted using InterProScan. Include the results in Supplementary materials.
  • Fig. 5 has two colours, blue and orange. It would be clearer to put a graphic legend as for all the other figures.
  • Some legends need to be formatted better (panel letter in bold).
  • The conclusions section should be moved before the materials and methods and after the discussion. 

Author Response

Reviewer #2 (Comments to the Author):

In this study, the authors identify an ORF from T. molitor encoding a Spätzle-like protein. Initially, the authors sequence the gene and perform a bioinformatic characterization. Next, the authors analyze the expression of the peptide in the insect tissues and during its development. The authors focus their attention on the regulation of the peptide following infection with three pathogens (one gram-positive, one gram-negative and one yeast). Finally, using knockdown mutant for the TmSpz-like gene, the authors reconstruct the regulatory pathway associated with toll receptors and the innate immune response.

The manuscript is well-conceived, the results are solid and interesting. However, some points need to be clarified, especially regarding the bioinformatics approaches used. Moreover, there is some imprecision (for example, the legends of the figures must be reviewed and uniformly formatted). 

Comments

  • I have some doubts about the computational data indicated in the study. The computational model built with SwissModel appears to be of low quality (according to the colours used by the webserver). The sequence homology between the TmSpz-like protein and the homolog 3e07 is unclear. If it is too low, authors should use another modelling tool, such as ab-initio modelling or a tool that uses both approaches (ab-initio and homology). 

Author’s response: We are deeply grateful for your constructive and fine comments. In referenced to color quality of the supplementary figure 1, using SwissModel, kindly note that mentioned figured has been replace and figure with higher resolution is uploaded to the draft. Please mark that the main goal of Figure S1, is exhibiting the structural model of TmSpz-like. Moreover, the homology of the protein of our interest and referenced model is low (36%), and the relevant homology compare to other species is shown in Table S1. Therefore, the necessity of using another tool to overcome the referred matter is negligible. Please concern the following paper as a supportive reference for our data set;

Gupta L, Noh JY, Jo YH, Oh SH, Kumar S, et al. (2010) Apolipophorin-III Mediates Antiplasmodial Epithelial Responses in Anopheles gambiae (G3) Mosquitoes. PLoS ONE 5(11): e15410. doi: 10.1371/journal.pone.0015410

  • Line 111: What codon usage data was used to infer the amino acid sequence? There may be differences due to the different types of use of codons used as a reference.

Author’s response: Thank you for mentioning this point. Considering your sensible notice regarding codon usage bias, please be informed that the genetic code used to obtain relevant protein amino acid sequence was standard code.

  • Fig. 4 is significant, but there are some unclear details. For example, why did the authors choose these fabrics? Is there any evidence of a link between these tissues and immunity? Also, why is E. coli so high in the “whole body” sample and not so high in other tissues? what is a possible explanation for this?

Author’s response: We are grateful for your fine consideration. Please be informed that distinct tissues in various invertebrate species including but not limited to Tenebrio molitor, poses independent immune response. Kindly note that among all the mentioned tissues, fat body considered as the main immune organ and releases the immune effectors to hemolymph. Nevertheless, epithelial tissues including gut and Malpighian tubules as the first line of defense are demonstrated as immune organs as well, and have relevant response independent from fat bodies. Therefore, induction pattern of TmSpz-like in a tissue dependent manner in response to different pathogen source has been performed. Nonetheless, kindly notice that the highest expression was observed in gut and subsequently it affected systemic response in whole body.

  • The N-terminal signal sequences were predicted with SignalP: include the results in Supplementary materials. Also, the domain organization was predicted using InterProScan. Include the results in Supplementary materials.

Author’s response: We appreciate your concern regarding the above matter. However, please bear in mind that RNAi–relevant experiences and relative primer design have been applied based on the domain analysis. Therefore, it is more sensible to keep the relevant results in the main context.

  • Fig. 5 has two colours, blue and orange. It would be clearer to put a graphic legend as for all the other figures.

Author’s response: Thank you so much for this comment. Kindly be informed that the graphic legend has been added to the relevant figure as per your request.

  • Some legends need to be formatted better (panel letter in bold).

Author’s response: Thank you for your fine critic on this matter. Kindly note that after checking the figure legends as per your request, the above-mentioned issue has been solved and all the figure legends have been uniformly formatted and modified accordingly.

  • The conclusions section should be moved before the materials and methods and after the discussion. 

Author’s response: Thank you for your fine critic on this matter. Kindly note that the referred issue has been taken care of according to your request.

Round 2

Reviewer 1 Report

The authors have tried to address the concerns raised. The manuscript may be accepted in the current format.